# Comparison of neural mobilization and conservative treatment on pain, range of motion, and disability in cervical radiculopathy: A randomized controlled trial

Shazia Rafiq[1]*, Hamayun Zafar[2,3,4,5], Syed Amir Gillani[6], Muhammad Sharif Waqas[7], Amna Zia[8], Sidrah Liaqat[8], Yasir Rafiq[9]

**1** Physiotherapy Department, Jinnah Hospital, Lahore, Pakistan, **2** University Institute Physical Therapy, University of Lahore, Lahore, Pakistan, **3** College of Applied Medical Sciences King Saud University, Riyadh, Saudia Arabia, **4** Rehabilitation Research Chair, King Saud University, Riyadh, Saudia Arabia, **5** Department of Odontology, Clinical Oral Physiology, Faculty of Medicine, Umea University, Umea, Sweden, **6** Allied Health Science, University of Lahore, Lahore, Pakistan, **7** Services Hospital, Lahore, Pakistan, **8** Physiotherapy Department, Mayo Hospital, Lahore, Pakistan, **9** Pathology Department, Combined Military Hospital, Kohat, Pakistan

* shazesarfraz@gmail.com

## Abstract

### Objective

The objective of the study was to compare the effectiveness of neural mobilization technique with conservative treatment on pain intensity, cervical range of motion, and disability.

### Methods

It was a randomized clinical trial; data was collected from Mayo Hospital, Lahore. Eighty-eight patients fulfilling the sample selection criteria were randomly assigned into group 1 (neural mobilization) and group 2 (conventional treatment). Pain intensity was measured on a numeric pain rating scale, range of motion with an inclinometer, and functional status with neck disability index (NDI). Data were analyzed using SPSS, repeated measure ANOVA for cervical ranges and the Friedman test for NPRS and NDI were used for within-group analysis. Independent samples t-test for cervical ranges and Mann-Whitney U test for NPRS and NDI were used for between-group comparisons.

### Results

There was a significant improvement in pain, disability, and cervical range of motion after the treatment in both groups compared to the pre-treatment status (p < 0.001), and when both groups were compared neural mobilization was more effective than conventional treatment in reducing pain and neck disability (p < 0.001), but there was no significant difference present in the mean score of cervical range of motion between both groups. (p>0.05).

**Data Availability Statement:** All relevant data are within the paper and its Supporting Information files.

**Funding:** The authors received no specific funding for this work.

**Competing interests:** The authors have declared that no competing interests exist.

## Conclusions

The present study concluded that both neural mobilization and conservative treatment were effective as an exercise program for patients with cervical radiculopathy, however, neural mobilization was more effective in reducing pain and neck disability in cervical radiculopathy.

## Trial registration

RCT20190325043109N1.

## Introduction

Cervical Radiculopathy (CR) is a disorder of the spinal nerve roots that are largely caused by a space-occupying lesion, disc herniation compression, and bony spur typically osteophytes in degenerating cervical spine which can lead to nerve root inflammation, impingement, or both [1]. These lesions can trigger pain receptors in the soft tissues and joints of the cervical spine that can lead to both sensory or motor changes in the upper extremity such as loss of or altered sensation, numbness and tingling in the upper extremity, muscular weakness in the arms, hands, neck or scapular region, and pain along the nerve pathways into the hand and arm, depending on affected nerve roots [2, 3]. The incidence of cervical radiculopathy annually is about 107.3 per 100,000 for males and 63.5 per 100, 000 for females. This incidence increases in the fifth decade of life up to 203 per 100 000 [4].

Among the Pakistani population, irregular physical activity, intensifying stress levels and deficiency of exercise are causing tremendous problems in daily routine [5]. These factors result in the bone, immune system, and muscle weakness as well as disturbance of normal body mechanics, resulting in instability of the spinal segments and related structures that aggravate the underlying musculoskeletal problems [6]. For establishing criteria for the diagnosis of cervical radiculopathy, different diagnostic tests are devised. MRI and EMG are important for diagnosing cervical radiculopathy [7].

Cervical radiculopathy can be treated surgically but there is a large number of evidence suggesting conservative management to be more effective than surgical treatment, suggesting multimodal treatment strategies that include cervical traction, manual therapy techniques, and strengthening exercises [8, 9].

Although there are limited high-quality evidences on the best nonoperative therapy for cervical radiculopathy, these are used to relieve discomfort and pain [8, 10, 11]. Neural mobilizations have been studied in various populations such as low back pain, carpal tunnel syndrome, lateral epicondylalgia, and cervicobrachial pain [12]. Neural mobilization is said to affect the axoplasmic flow, movement of the nerve and its connective tissue and the circulation of the nerve by alteration of the pressure in the nervous system and dispersion of intraneural edema. Nerve gliding exercises are a sequence of the positioning of the upper limb or lower limb to elongate nerves used in the treatment of multiple musculoskeletal disorders. However further research is required for validation of this neural mobilization concept, particularly in cervical radiculopathy [10, 13].

Many studies have been done on the treatment of cervical radiculopathy. But most of them are inconclusive in terms of defining appropriate treatment options that would be efficacious for the treatment of this pathology [9].

The purpose of this study was to assess the neural mobilization technique as an effective treatment to improve neck mobility, and reduce pain intensity and disability for cervical

radiculopathy through appropriate randomized controlled trials, taking the factors and outcome measures under consideration that are not addressed properly in previous literature.

## Material and methods

This was a parallel-group randomized controlled trial with a 1:1 allocation ratio into two groups. The study was conducted at the physical therapy department of Mayo Hospital Lahore. This randomized trial was conducted according to consolidated standards of reporting trial CONSORT guidelines (2010) [14] from July 2019 to July 2020 as mentioned in Fig 1. Ethical approval was gained from the University of Lahore (Ref#IRB-UOL-FAHS/373-VI/2018). Written informed contents in Urdu or English were taken from all the subjects to participate in the

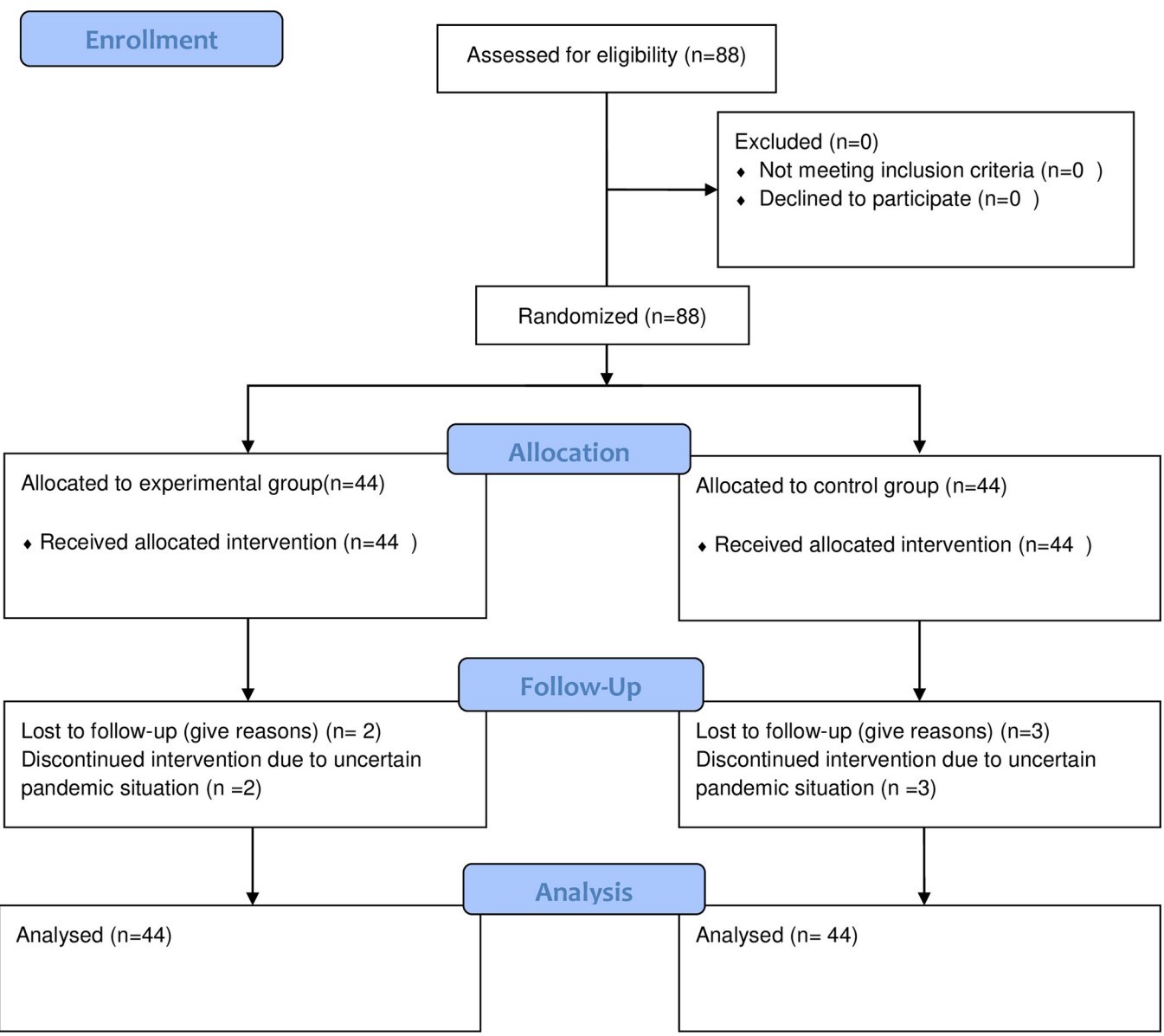

**Fig 1. Consort flow diagram.**

study. The trial was prospectively registered through the Iranian Registry of Clinical Trials (IRCT20190325043109N1) on 30/6/2019. Total 88patients (44 in Group 1 and 44 in group 2) were included through the convenience sampling technique in this study, using an effect size of 0.70, using (mean ±SD) in group 1(neural mobilization group) (3.35±1.49) and group 2 (exercise group) (4.45±1.63), at the level of significance 0.05 and power 9 0% [15]. To recruit the patients in this study, patients between ages 35–50 years, of both genders, having radiating symptoms of cervical radiculopathy from at least 2 months and not more than 6 months (for the diagnosis of the patients with cervical radiculopathy clinical predictor rule was applied consisting of Spurling's test, Distraction test, upper limb nerve tension test ULNTT for the median nerve and ipsilateral neck rotation), with no previous cervical surgeries, no loss of the upper limb movement and willing to participate were chosen. Subjects having traumatic history, osteoporosis, hypermobility, circulatory disturbances, tumor-causing cervical radiculopathy and who were not willing were excluded from the study. Patients were asked to sign a consent form and give their will regarding being enrolled in the study.

## Randomization and concealment of allocation

The randomization sequence was created by using Excel 2016 with a 1:1 allocation using simple randomization by an independent researcher who was not participating in the treatment of patients.

Patients were allocated to two groups by concealment of allocation (sealed envelopes).

Allocation concealment was achieved with sequentially numbered, opaque, sealed, envelopes SNOSE. SNOSE was used according to the guidelines of Doig and Simpson [16].

An independent researcher with no clinical involvement in the trial made the concealed envelopes. 88 Envelopes were made. Half envelopes contained folded papers with Treatment A (group 1) written on them and the remaining half contained folded papers with Treatment B (group 2) written on them. A unique randomized number was allocated to these envelopes and shuffled vigorously. Then envelopes were arranged sequentially and handed over to another independent researcher. The assessor pre-tested the participant and to eligible subjects, the envelope was allocated to the subject. The therapist recorded the information on the envelope and opened it afterward to maintain the concealment. The assessor recorded the post-treatment findings and another independent analyst analyzed the data.

## Blinding

In this study patients, assessors, and data analysts were blinded to the allocation of treatment groups in this study. Except for the therapist, all other staff was kept blinded as they were not informed about the details of allocation. The trial adhered to established procedures to maintain separation between the staff who was collecting outcome measurements and the therapist. Patients were blinded to treatment allocation as treatment was given in separate rooms for each group. A therapist who is not blinded did not take the outcome measurements. All the other assessors, investigators, and analysts did not know the details of treatment.

## Intervention

All the subjects were given hot packs for 10 minutes before the treatment in both groups.

Subjects fulfilling the sample selection criteria were given treatment for 12 sessions (3 times per week for 4 weeks). Pre-assessment was done at baseline, the second assessment was done after 2 weeks and the final post-assessment was done at the end of the 12th session in the 4th

week in both groups. Fig 1 has provided a detailed description of the participant's flow through the study.

## Group 1

In this group neural mobilization technique with sliding of the median nerve was applied in 1 set of 10 repetitions with 3 seconds hold in each repetition. Neural mobilization was done according to the technique described by David Butler [16]. The subject was placed in supine position with the shoulder in abduction and lateral rotation; elbow in extension, forearm in supination and wrist, finger and thumb in extension position then finally, to apply the stretch shoulder was taken in greater abduction and cervical spine in contralateral side flexion.

In this group conservative treatment was also given which included 3 sets of cervical isometrics exercises with 10 repetitions in each direction with 5 seconds hold in each isometric movement and 30 seconds rest period between each set. Isometric exercises were performed with the patient in a sitting position. For the cervical flexor isometrics, the therapist placed his hand on the anterior side of the patient's forehead, and the patient was instructed to push his head against the therapist's hand in the forward direction, for the cervical extensor isometrics, the therapist placed his hand on the posterior side of the patient's head and was instructed to push the head in the backward direction, for the cervical side flexors therapist placed his hand on the lateral sides (both right and left) of patient's head and the patient was instructed to push sideways and for the cervical rotator, the isometrics therapist placed his hand on the lateral side of the patient's head(both right and left) and instructed him to rotate the head.

## Group 2

In group 2 conservative treatments were given which included cervical isometrics exercises with 10 repetitions in each direction, with 5 seconds hold in each isometric movement, 3 sets of these exercises were performed with the rest period of 30 seconds.

Isometric exercises were performed with the patient in a sitting position with a similar procedure mentioned in group 1

## Outcome measures

The main outcome was to measure the effectiveness of the neural mobilization technique on pain intensity measured on NPRS, range of motion measured on inclinometer, and measuring the effects of treatment on quality of life measured through Neck Disability Index (NDI).

The numeric pain rating scale (NPRS) is an 11point scale ranging from 0 to 10, and a higher score indicates greater intensity of pain. 0 stands for no pain and 10 for worst possible pain. In subjects with neck pain, NPRS has ICC = 0.67 [17]. The inclinometer is a device used for measuring angles and the cervical range of motion Inclinometer has ICC = 0.85 [18].

NDI is a self-reported questionnaire used for measuring functional status in subjects with neck pain, the questionnaire contains a total of 10 sections and for each section, the total possible score is 5: if the first statement is marked the section score = 0 if the last statement is marked it = 5. If all ten sections are completed the score is calculated by the following formula: Score: /50 Transform to percentage score x 100 = %points.

The lesser score represents a lesser disability or better functional status, and NDI has fair test-retest reliability [19].

## Baseline measurements

Patient age, gender, baseline pain, ROM, and quality of life were noted at the time of recruitment.

**Table 1. Baseline measurement of pain (NPRS) and disability (NDI).**

| Variable | At Baseline | | | |
|---|---|---|---|---|
| | Neural mobilization group | Conservative treatment group | Z score | p-value |
| | Mean rank | Mean rank | | |
| NPR | 44.89 | 44.11 | -0.14 | 0.88 |
| NDI | 42.84 | 46.16 | -0.61 | 0.54 |

## Data analysis

Data were analyzed using SPSS version 21. Descriptive analyses (mean, variance, standard deviation) were performed for quantitative data. Frequencies and percentages were calculated for categorical and nominal data of gender. Data were analyzed for normality by applying Shapiro- Wilk test. To compare the variables between both groups, independent samples t-test was used for cervical ranges and Mann-Whitney U test was used for NPRS and NDI. To compare the variable within groups, repeated measure ANOVA was used for cervical ranges and the Friedman test was used for NPRS and NDI. P-value $\leq 0.05$ was considered significant. Intention to treat analysis with the technique of last observation carried forward (LOCF) was used to handle the missing data due to loss of follow-up.

## Results

Total of 88 subjects were included in this study, 5 patients (2 from group 1 and 3 from group 2) were lost to follow-up, mostly due to the ongoing pandemic situation, and missing data were managed through intention to treat analysis with the technique of last observation carried forward. The mean age of the subject in group 1 was 41.09± 6.05in and group 2 was 42.22 ± 5.62. According to the gender distribution in the neural mobilization group, 15 (34.1%) were males and 29 (65.9%) were females and in the conventional treatment group, 13 (29.5%) were males and 31 (70.5%) were females.

Shapiro-Wilk test of normality has shown that p values were greater than 0.05 for cervical range of motion variables and less than 0.05 for neck disability index (NDI) and numeric pain rating scale (NPRS), showing that data was normally distributed for cervical range of motion (ROM) and not normally distributed for NDI and NPRS. Pretreatment comparison of variables (NPRS, NDI, and cervical ROM) between both groups showed that there was no statistically significant difference between both groups at the baseline as presented in Tables 1 and 2.

Comparison of variable with in group 1 has shown that there was a statistically significant difference between pre, mid, and post-treatment NPRS scores. $X^2 = 82.14$, p<0.001. Post hoc analysis with Wilcoxon signed-rank was conducted with Bonferroni correction applied,

**Table 2. Baseline measurement of cervical range of motion (ROM).**

| Variable | At baseline | | | |
|---|---|---|---|---|
| | Neural mobilization group | Conservative treatment group | Mean Change (95% CI) | p-value |
| | Mean ± SD. | Mean ± SD. | | |
| Cervical flexion | 39.09± 9.41 | 38.54± 11.48 | 0.55(-3.90,4.99) | 0.80 |
| Cervical extension | 39.06± 12.26 | 42.04± 12.23 | 2.97(-8.16,2.21) | 0.25 |
| Cervical right side flexion | 29.81± 8.46 | 30.81± 10.21 | 1.00(-4.97, 2.97) | 0.61 |
| Cervical left side flexion | 29.88± 8.82 | 30.36± 8.55 | 0.47(-4.16, 3.20) | 0.79 |
| Cervical right rotation | 36.45± 10.36 | 38.38± 9.58 | 1.93(-6.16, 2.29) | 0.36 |
| Cervical left rotation | 40.09± 9.09 | 40.90± 10.50 | 0.81(-4.98, 3.34) | 0.69 |

resulting in a significant level set at p < 0.017. The Median (interquartile range (IQR)) for pre-treatment the in experimental group NPRS score was 6(5 to 6), mid-treatment was 4 (3 to 5) and post-treatment was 3 (2 to 4). There was a significant difference between pretreatment and mid-treatment ($Z = -5.76$, $p<0.001$), mid-treatment and post-treatment ($Z = -5.46$, $p<0.001$), and pretreatment and post-treatment ($Z = -5.74$, $p<0.001$), showing that NPRS score was significantly improved after 2 weeks and further improved after 4 weeks of treatment in the experimental group as shown in Table 3.

There was a statistically significant difference between pre, mid, and post-treatment NDI score.

$X^2 = 71.302$, $p<0.001$. Post hoc analysis with Wilcoxon signed-rank was conducted with Bonferroni correction applied, resulting in a significant level set at p < 0.017. The Median (IQR) for pretreatment Neck Disability Score (NDI) in the experimental group was 38(30 to 46), mid-treatment was 24 (20 to 28) and post-treatment was 14 (8.5 to 20). There was a significant difference between pretreatment and mid-treatment ($Z = -5.41$, $p<0.001$), mid-treatment and post-treatment ($Z = -5.68$, $p<0.001$), and pretreatment and post-treatment ($Z = -5.58$, $p<0.001$), showing that disability was significantly reduced after 2 weeks and further reduce after 4 weeks of treatment in the experimental group.

Pre and post-treatment comparison of cervical ranges of motion score in experimental has shown that post-treatment mean and standard deviation improved with $p < 0.05$, showing that neural mobilization is effective in improving cervical ranges in patients with cervical radiculopathy as shown in Table 3.

Comparison of variables within the group 2 has shown that there was a statistically significant difference between pre, mid, and post-treatment NPRS scores. $X^2 = 71.02$, $p = 0.00$. Post hoc analysis with Wilcoxon signed-rank was conducted with Bonferroni correction applied, resulting in a significant level set at p < 0.017. Median (IQR) for pretreatment in the control group NPRS score was 6(5 to 6), mid-treatment was 5 (3.25 to 6) and post-treatment was 4 (2.25 to 5). There was a significant difference between pretreatment and mid-treatment

**Table 3. Comparison of pain (NPRS), disability (NDI), and cervical mobility (ROM) within the experimental group.**

| Variable | Baseline | 2nd week follow up | At the end of the 4th week | $X^2$ / F | p-Value |
|---|---|---|---|---|---|
| NPRS Median (IQR) | 6(5 to 6) | 4 (3 to 5) | 3 (2 to 4) | $X^2 = 82.14$ | < 0.001* |
| NDI Median (IQR) | 38(30 to 46) | 24 (20 to 28) | 14 (8.5 to 20) | $X^2 = 71.30$ | < 0.001* |
| Cervical flexion Mean ± S.D | 39.09± 9.41 | 44.65 ±8.30 | 48.22± 8.89 | F = 78.94 | 0.01** |
| Cervical extension Mean ± S.D | 39.06± 12.26 | 45.65 ±12.74 | 49.56 ±13.09 | F = 66.69 | < 0.001 |
| Cervical right side flexion Mean ± S.D | 29.81± 8.46 | 34.29 ±8.84 | 37.93 ±9.06 | F = 153.99 | < 0.001 |
| Cervical left side flexion Mean ± S.D | 29.88± 8.82 | 35.18± 9.05 | 39.31± 9.10 | F = 132.76 | < 0.001 |
| Cervical right rotation Mean ± S.D | 36.45± 10.36 | 43.02 ±10.12 | 46.88 ±10.21 | F = 75.52 | < 0.001 |
| Cervical left rotation Mean ± S.D | 40.09± 9.09 | 45.38± 9.39 | 49.06 ±9.45 | F = 108.85 | < 0.001 |

* Friedman test

** Repeated measure ANOVA (for all cervical ranges)

**Table 4. Comparison of pain (NPRS), disability (NDI), and cervical mobility (ROM) within the control group.**

| Variable | Baseline | 2nd week follow up | At the end of the 4th week | X² / F | p-Value |
|---|---|---|---|---|---|
| NPRS | 6(5 to 6) | 5 (3.25 to 6) | 4 (2.25 to 5) | $X^2 = 71.02$ | < 0.001* |
| NDI | 40 (30 to 49) | 30 (22 to 38) | 22 (16 to 30) | $X^2 = 72.11$ | < 0.001* |
| Cervical flexion | 38.54± 11.48 | 44.06 ±8.74 | 47.63 ±7.62 | F = 54.28 | < 0.001** |
| Cervical extension | 42.04± 12.23 | 48.09 ±9.20 | 50.27 ±8.79 | F = 32.17 | < 0.001 |
| Cervical right side flexion | 30.81± 10.21 | 33.77 ±8.04 | 36.31 ±7.69 | F = 25.78 | < 0.001 |
| Cervical left side flexion | 30.36± 8.55 | 34.09± 6.65 | 36.38± 6.92 | F = 33.60 | < 0.001 |
| Cervical right rotation | 38.38± 9.58 | 42.70 ±9.64 | 45.54 ±9.97 | F = 54.97 | < 0.001 |
| Cervical left rotation | 40.90± 10.50 | 43.86± 9.61 | 45.88 ±9.44 | F = 33.55 | < 0.001 |

*Friedman test

** Repeated measure ANOVA

A comparison of the mean and standard deviation of NPRS between group 1 and group 2 has shown that there was no significant difference in NPRS score at baseline, as value p>.05, but there was significant difference after 2nd and further improvement after 4th week, as value p < .05, showing that neural mobilization is more effective in reducing pain.

(Z = - 5.38, p = 0.00), mid-treatment and post-treatment (Z = -4.74, p<0.001), and pretreatment and post-treatment (Z = -5.49, p<0.001), showing that NPRS score was significantly improved after 2 weeks and further improved after 4 weeks of treatment in the control group.

There was a statistically significant difference between pre, mid, and post-treatment NDI scores, $X^2 = 72.11$, p<0.001. Post hoc analysis with Wilcoxon signed-rank was conducted with Bonferroni correction applied, resulting in a significant level set at p < 0.017. The Median (IQR) for pretreatment Neck Disability Score (NDI) in the control group was 40(30 to 49), mid-treatment was 30 (22 to 38) and post-treatment was 22 (16 to 30). There was a significant difference between pretreatment and mid-treatment (Z = -5.23, p<0.001), mid-treatment and post-treatment (Z = -5.26, p<0.001), and pretreatment and post-treatment (Z = -5.51, p<0.001), showing that disability was significantly reduced after 2 weeks and further reduce after 4 weeks of treatment in the control group.

Pre and post-treatment comparison of cervical ranges of motion score in group 2 has shown that post-treatment mean and standard deviation improved with p < 0.05, showing that conventional treatment is effective in improving cervical ranges in patients with cervical radiculopathy as shown in Table 4.

A comparison of NDI between group 1 and group 2 has shown that there was no significant difference in NDI score at baseline and even after 2 weeks of treatment, as value p>.05, but there was a significant difference after 4 weeks in the neural mobilization group as the mean rank was 35.30 with p<0.05 as shown in Table 5.

Comparison of the mean and standard deviation of cervical range of motion between group 1 and group 2 has shown, that there was no significant difference in cervical ranges at baseline, after 2nd and 4th weeks of treatment, as p-value >.05, so experimental and control groups showed equal improvement as shown in Table 6.

**Table 5. Comparison of NPRS and NDI between the experimental and control group.**

| Variable | At Baseline | | | | At the end of 4th week | | | |
|---|---|---|---|---|---|---|---|---|
| | Neural mobilization group | Conservative treatment group | Z score | p-value | Neural mobilization group | Conservative treatment group | Z score | p-value |
| | Mean rank | Mean rank | | | Mean rank | Mean rank | | |
| NPRS | 44.89 | 44.11 | -0.14 | 0.88 | 37.45 | 51.55 | -2.63 | 0.008 |
| NDI | 42.84 | 46.16 | -0.61 | 0.54 | 35.30 | 53.70 | -3.38 | 0.001 |

**Table 6. Comparison of cervical range of motion between experimental and control group.**

| Variables | At baseline | | | | At the end of 4th week | | | |
|---|---|---|---|---|---|---|---|---|
| | Neural mobilization group | Conservative treatment group | Mean Change (95% CI) | p-value | Neural mobilization group | Conservati ve treatment group | Mean Change (95%CI) | p-value |
| | Mean±SD. | Mean±SD. | | | Mean±SD. | Mean±SD. | | |
| Cervical flexion | 39.09± 9.41 | 38.54± 11.48 | 0.55(-3.90,4.99) | 0.80 | 48.22± 8.89 | 47.63± 7.62 | 0.59 (-2.92, 4.10) | 0.73 |
| Cervical extension | 39.06± 12.26 | 42.04± 12.23 | 2.97(-8.16,2.21) | 0.25 | 49.56 ±13.09 | 50.27 ±8.79 | 0.70(-5.43, 4.02) | 0.76 |
| Cervical right side flexion | 29.81± 8.46 | 30.81± 10.21 | 1.00(-4.97, 2.97) | 0.61 | 37.93 ±9.06 | 36.31 ±7.69 | 1.61(-1.95 to 5.17) | 0.37 |
| Cervical left side flexion | 29.88± 8.82 | 30.36± 8.55 | 0.47(-4.16, 3.20) | 0.79 | 39.31± 9.10 | 36.38± 6.92 | 2.93(-0.46 to 6.35) | 0.93 |
| Cervical right rotation | 36.45± 10.36 | 38.38± 9.58 | 1.93(-6.16, 2.29) | 0.36 | 46.88 ±10.21 | 45.54 ±9.97 | 1.34(-2.93 to 5.61) | 0.53 |
| Cervical left rotation | 40.09± 9.09 | 40.90± 10.50 | 0.81(-4.98, 3.34) | 0.69 | 49.06 ±9.45 | 45.88 ±9.44 | 3.18(-0.82 to 7.18) | 0.11 |

## Discussion

Cervical radiculopathy is a clinical syndrome, that often affected persons are unable to perform their social obligations, do physical tasks pertinently, and lose working hours [20]. In literature for cervical radiculopathy many non-surgical treatment options are discussed and multimodal conservative approach has been proven to be more effective in improving the symptoms [21].

Subjects with cervical radiculopathy show deconditioning of cervical muscles due to inactivity. Exercises are shown to be beneficial in improving well-being of a person and reducing disability. These has also shown to improve sleep, emotional and physical functioning, cognitive functioning, and reducing depression or anxiety [22]. In present study neural mobilization was used along with strengthening exercises of cervical muscles. This showed results in consistent with Liang et al, as it showed improvement in range of motion of cervical spine and reduction in pain [10]. Past studies show the moderate benefit of these exercises in reducing pain and improving strength in patients with cervical radiculopathy [2]. Exercises promote analgesic effect in musculoskeletal pain, as shown in the study of Lima et al exercises can reduce the hyperalgesia in muscle pain or after an injury. These results are corelate with the present study as reduction in pain is observed in both groups [23]. This exercise-induced hypoalgesia or analgesia predicts greater pain relief and improvement in cervical functioning by restoring muscular balance through strengthening cervical muscle exercises, this in turn impacts the quality of life positively, enhancing independence and reducing disability [22]. In the present study, comparison of NDI score between group 1 and group 2 has shown that there was significant improvement ($p<0.05$) in NDI score in group 1 with neuro-mobilization while the comparison of NPRS between group 1 and group 2 has also shown there was a significant difference after 2nd and further improved after 4th week, as value $p < .05$, showing that neural mobilization is more effective in reducing pain and improving functional status in cervical

radiculopathy. Many other studies also support these results as exercise intervention containing isometric exercise of deep neck flexor muscles showed alleviation in levels of pain and disability, measured on outcome scale of numeric pain rating scale NPRS and neck disability index NDI respectively [24–27]. In present study both groups were given hot fermentation along with exercise regime. This treatment program when combined with heating modalities, shows decreased pain in patients with cervical radiculopathy as shown in a randomized controlled trial of Diab et al. [28].

In addition to strengthening exercises, neurophysiological and analgesic effect of neural mobilization also predicts relief in pain and improvement in cervical functioning. Mechanism of pain reduction is by stimulating mechanical receptors, reducing edema and improving circulation [29]. Results of present study also correlate with a study conducted by Boyle et al. and Beniciuk et al in which the effects of manual physical therapy including neural mobilization were assessed on cervical radiculopathy. These studies concluded that mechanistic effects of neural mobilization differ from sham treatment, it resulted in neurophysiological effect and hypoalgesia of C-mediated fibers and there was a reduction in sensory descriptors. Results showed improvement in range of cervical motion, alleviation in cervical pain, improvement in disability, and enhancing functional capacity of the patients. Significant improvements were shown on NDI and NPRS [30], which are consistent with the present study as pre and post-treatment comparison of NDI scores in group 1 has shown that neural mobilization is effective in improving functional status in patients of cervical radiculopathy [31].

Results of present study are also in consistent with study of Dong-Gyu et al. which compared effects of neural-mobilization with manual cervical traction on pain, disability, muscle endurance and range of motion in individuals with cervical radiculopathy. As in present study these outcomes were improved in group treated with neural-mobilization along with cervical isometric exercises [13].

In a randomized trial of Kim et al, the patient showed improvement in NPRS, NDI, range of motion and endurance of deep flexor cervical muscles in patients with cervical radiculopathy [13]. Results of the presents study have shown that cervical ranges were significantly improved in both groups (p<0.05) but a comparison of cervical range of motion between group 1 and group 2 has shown, that there was no significant difference in cervical ranges after 4 weeks of treatment, as p-value >.05, so experimental and control groups showed almost equal improvement.

From the literature, it seems that multimodal management interventions are more effective than uni-modal strategies. Among non-surgical conservative multimodal management comprised of neurodynamic mobilization and exercises are more effective as conservative treatment in participants with cervical radiculopathy [8, 32].

In present study an effort has been made to consider all the elements of high-quality evidence so knowledge base could be improved about the effective treatment options for cervical radiculopathy. However, further research is required focusing on reducing the variability in patient selection in clinical trials, which will further optimize clinical practice. Therefore, there are a few limitations in this study that may affect the results and should be considered in future studies. First, when given combined treatment it is often times difficult to interpret the result of a single intervention. Second, subjective or objective measurement of upper limb pain and numbness due to radiculopathy were not included. Third, the sample was collected only from one clinical setting so results cannot be generalized. Fourth, this study is biased in using neural mobilization technique on median nerve, therefore this technique may not be useful for participants with cervical radiculopathy at other cervical spinal levels. Fifth, long term effects should be evaluated in future studies.

## Conclusion

The present study concludes that neural mobilization combined with cervical isometrics is more effective as treatment program for patients with cervical radiculopathy in reducing pain, increasing cervical range of motion, and reducing neck disability, than cervical isometric exercises alone.

## Implication

Being cost-effective treatment technique and non-availability of different expensive modalities in most clinical settings, this study will help many people with cervical radiculopathy and can enhance the scope of rehabilitation in these subjects.

## Supporting information

**S1 Checklist. Consolidated standards of reporting trials checklist.**
(DOCX)

**S1 File. Study protocol.**
(DOCX)

**S2 File. Synopsis.**
(DOCX)

**S1 Data. Study data.**
(XLSX)

## Acknowledgments

I would like to express my deep gratitude to Professor Humayun Zafar, my research supervisor, for their patient guidance, enthusiastic encouragement, and useful critiques of this research work.

## Author Contributions

**Conceptualization:** Shazia Rafiq, Hamayun Zafar.

**Data curation:** Shazia Rafiq, Sidrah Liaqat.

**Formal analysis:** Amna Zia.

**Methodology:** Shazia Rafiq, Amna Zia.

**Resources:** Shazia Rafiq.

**Supervision:** Hamayun Zafar.

**Validation:** Syed Amir Gillani, Muhammad Sharif Waqas, Yasir Rafiq.

**Visualization:** Syed Amir Gillani, Muhammad Sharif Waqas, Yasir Rafiq.

**Writing – original draft:** Amna Zia, Sidrah Liaqat.

**Writing – review & editing:** Amna Zia, Sidrah Liaqat.

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
