## [Decision Letter · Decision Letter 0]

5 Oct 2022

PONE-D-22-23496Effectiveness of Neural Mobilization on Pain, Range of motion, and Disability in Cervical Radiculopathy: A Randomized Controlled TrialPLOS ONE

Dear Dr. Rafiq,

Thank you for submitting your manuscript to PLOS ONE. After careful consideration, we feel that it has merit but does not fully meet PLOS ONE’s publication criteria as it currently stands. Therefore, we invite you to submit a revised version of the manuscript that addresses the points raised during the review process.

We look forward to receiving your revised manuscript.

Kind regards,

Walid Kamal Abdelbasset, Ph.D.

Academic Editor

PLOS ONE

Journal Requirements:

2. We noticed you have some minor occurrence of overlapping text with the following previous publication, which needs to be addressed:

https://www.sciencedirect.com/science/article/abs/pii/S1521694215000297?via%3Dihub

In your revision ensure you cite all your sources (including your own works), and quote or rephrase any duplicated text outside the methods section. Further consideration is dependent on these concerns being addressed.

"unfunded study"

5. We noted in your submission details that a portion of your manuscript may have been presented or published elsewhere. Please clarify whether this [conference proceeding or publication] was peer-reviewed and formally published. If this work was previously peer-reviewed and published, in the cover letter please provide the reason that this work does not constitute dual publication and should be included in the current manuscript.

Reviewers' comments:

Reviewer's Responses to Questions

**Comments to the Author**

1. Is the manuscript technically sound, and do the data support the conclusions?

Reviewer #1: Yes

Reviewer #2: Partly

2. Has the statistical analysis been performed appropriately and rigorously? 

Reviewer #1: Yes

Reviewer #2: I Don't Know

3. Have the authors made all data underlying the findings in their manuscript fully available?

Reviewer #1: Yes

Reviewer #2: Yes

4. Is the manuscript presented in an intelligible fashion and written in standard English?

Reviewer #1: Yes

Reviewer #2: Yes

5. Review Comments to the Author

Reviewer #1: Please make the following modifications to the paper:

1. Please revise the study title in light of the current study's objectives.

2. Work on balancing the different sections of the research abstract, as the method and results sections are clearly overly detailed.

3. I hope the author(s) will be as brief as possible in these two sections of the abstract - methods and results - with a focus on the most important research findings in the results section of the study abstract.

4. The introduction is far too long and detailed. I hope to limit the introduction to two or three paragraphs at most, with the final paragraph devoted to the current research problem and how to solve it within the framework of the current study's design objective.

5. I noticed that the material and method sections were written as paragraphs with side headings. The names of the side headings, in my opinion, are ineffective in describing what is related to them in this section, which the author(s) should change.

6. Please relate Figure 1 to the texts in the material and methods section.

7. The author(s) should carefully review the progress of their research results to avoid the presence of statistical flaws, some of which were discovered while reviewing the paper, with the goal of displaying graphic images to alleviate the boredom caused by displaying the results only in the style of statistical tables.

8. Author(s) should also avoid using side titles in the results section, which is not recommended.

9. I noticed that the discussion section served as an introduction to the study, and the discussion appeared to be completely inadequate in terms of proving or denying the current study's findings with previous studies, so please review this shortcoming.

10. In the final paragraph of the discussion, consider emphasizing the study's future trends, as well as its strengths and weaknesses.

11. Rewrite the conclusion section in the context of resolving the research problem and clarifying the study's goal achievement.

12. Updating the reference list and replacing outdated references prior to 2015 with new ones.

13. In general, the paper requires linguistic correction to eliminate grammatical errors and typos.

//Good luck//

Reviewer #2: In the first comment, all pages have no number and no line numbers so it is very difficult to detect the pages in the consort checklist and lines in the manuscript

This manuscript requires a significant amount of improvement, the authors have designed a randomized controlled study to show the effectiveness of the neural mobilization technique on cervical radiculopathy which has already been studied more times in the past by other researchers as the author wrote in the last paragraph of introduction,

I already read reference No 9 in the last paragraph which is completely against the fact you write (Liang, Long PhDa,b; Feng, Minshan PhDa,b; Cui, Xin MSa; Zhou, Shuaiqi MSa,b; Yin, Xunlu PhDa,b; Wang, Xingyu MDc; Yang, Mao MDc; Liu, Cunhuan MDd; Xie, Rong MDa; Zhu, Liguo PhDa,b,∗; Yu, Jie PhDa,b,∗; Wei, ,

Introduction

Need editing and rewrite

Methods:

What is the novelty of your method and measurement?

The treatment time was very short without follow up

The arrangement of the method is very weak and you should follow the consort checklist

Statistics

Completely Not clear –please rewrite again and discuss which test you used and why

all tables need to rearrange

Outcomes and estimation need to be explained well

Discussion:

needs to be as per well defined objectives

Describe sources of potential bias and imprecision

It has to be framed in such a way that readers are able to have good understanding of the current evidences and rationale of the paper

6. PLOS authors have the option to publish the peer review history of their article (what does this mean?). If published, this will include your full peer review and any attached files.

Reviewer #1: No

Reviewer #2: No

---

## [Author Response · Author response to Decision Letter 0]

24 Oct 2022

I am grateful to the reviewers for their insightful comments on the paper. We have been able to incorporate changes to reflect most of the suggestion provided by the reviewers. Here is a point-by-point response to the reviewer’s comments and concerns. 

Guidelines has been followed during development of manuscripts. 

2. We noticed you have some minor occurrence of overlapping text with the following previous publication, which needs to be addressed:

Reference of the article has been given in methodology. 

No specific funding was received for this research. 

All relevant data is available within the manuscript. Data cannot be shared publicly because of ethical concerns. Patients were included on a no objection base to conduct the study. Patients were not asked for permission to publish full encrypted data. 

Data availability statement has been updated on the cover page

5. We noted in your submission details that a portion of your manuscript may have been presented or published elsewhere. Please clarify whether this [conference proceeding or publication] was peer-reviewed and formally published. If this work was previously peer-reviewed and published, in the cover letter please provide the reason that this work does not constitute dual publication and should be included in the current manuscript.

This article is present in the pre-prints and was mentioned in the cover letter and link is as follow https://www.researchsquare.com/article/rs-223383/v1 . One other article containing different variables were extracted from this research and is published recently in BioMed journal; link is as follow https://www.hindawi.com/journals/bmri/2022/9385459

Remove it from abstract and check in methodology

Reviewers' comments:

Reviewer's Responses to Questions

Comments to the Author

1. Is the manuscript technically sound, and do the data support the conclusions?

Reviewer #1: Yes

Reviewer #2: Partly

2. Has the statistical analysis been performed appropriately and rigorously?

Reviewer #1: Yes

Reviewer #2: I Don't Know

3. Have the authors made all data underlying the findings in their manuscript fully available?

Reviewer #1: Yes

Reviewer #2: Yes

4. Is the manuscript presented in an intelligible fashion and written in standard English?

Reviewer #1: Yes

Reviewer #2: Yes

5. Review Comments to the Author

Reviewer #1: Please make the following modifications to the paper:

1. Please revise the study title in light of the current study's objectives.

Study title has been revised.

2. Work on balancing the different sections of the research abstract, as the method and results sections are clearly overly detailed.

Method and results sections are revised. 

3. I hope the author(s) will be as brief as possible in these two sections of the abstract - methods and results - with a focus on the most important research findings in the results section of the study abstract.

Method and results sections in abstract are revised. 

4. The introduction is far too long and detailed. I hope to limit the introduction to two or three paragraphs at most, with the final paragraph devoted to the current research problem and how to solve it within the framework of the current study's design objective.

Information in introduction section has been curtailed. 

5. I noticed that the material and method sections were written as paragraphs with side headings. The names of the side headings, in my opinion, are ineffective in describing what is related to them in this section, which the author(s) should change.

Methodology of other published articles in PLOS one is carefully reviewed. In this article subheadings are given where necessary to describe details of methodology. 

6. Please relate Figure 1 to the texts in the material and methods section.

Figure 1 has been mentioned in the text of material and methods. 

7. The author(s) should carefully review the progress of their research results to avoid the presence of statistical flaws, some of which were discovered while reviewing the paper, with the goal of displaying graphic images to alleviate the boredom caused by displaying the results only in the style of statistical tables.

I have carefully reviewed the result section but were unable to find the statistical flaws. It would be of great if you could point out what kind of flaws has been noticed. 

8. Author(s) should also avoid using side titles in the results section, which is not recommended.

Side titles in the results section has been removed. 

9. I noticed that the discussion section served as an introduction to the study, and the discussion appeared to be completely inadequate in terms of proving or denying the current study's findings with previous studies, so please review this shortcoming.

Discussion has been rewritten and effort has been made to rectify all shortcomings. 

10. In the final paragraph of the discussion, consider emphasizing the study's future trends, as well as its strengths and weaknesses.

Limitations, strength and weaknesses of the study are mentioned at end of discussion section. 

11. Rewrite the conclusion section in the context of resolving the research problem and clarifying the study's goal achievement.

Conclusion has been rewritten. 

12. Updating the reference list and replacing outdated references prior to 2015 with new ones.

New references have been added 

Reviewer #2: In the first comment, all pages have no number and no line numbers so it is very difficult to detect the pages in the consort checklist and lines in the manuscript

This manuscript requires a significant amount of improvement, the authors have designed a randomized controlled study to show the effectiveness of the neural mobilization technique on cervical radiculopathy which has already been studied more times in the past by other researchers as the author wrote in the last paragraph of introduction,

I already read reference No 9 in the last paragraph which is completely against the fact you write (Liang, Long PhDa,b; Feng, Minshan PhDa,b; Cui, Xin MSa; Zhou, Shuaiqi MSa,b; Yin, Xunlu PhDa,b; Wang, Xingyu MDc; Yang, Mao MDc; Liu, Cunhuan MDd; Xie, Rong MDa; Zhu, Liguo PhDa,b,∗; Yu, Jie PhDa,b,∗; Wei, ,

Study of Liang et al. concluded that Exercise alone or exercise plus other treatment may be helpful to patients with Cervical radiculopathy. However, exercise option should be carefully considered for each patient with CR. Large-scale studies using proper methodology are recommended that is why in our study carefully designed exercise plan was incorporated in both treatment groups and neural mobilization was additionally added in group 1 to measure its effectiveness. Result of our studies corelate with this study as both groups shows the significant improvement but adding neural mobilization was more helpful in improving pain and functional status. 

Introduction

Need editing and rewrite

Introduction section has been edited.

Methods:

What is the novelty of your method and measurement?

The treatment time was very short without follow up

This has been mentioned in the limitations of study.

The arrangement of the method is very weak and you should follow the consort checklist

Statistics

Completely Not clear –please rewrite again and discuss which test you used and why

all tables need to rearrange

Outcomes and estimation need to be explained well

Description of the test used for the analysis is mentioned under the data analysis heading and is as follow: 

Normality testing has shown that data were normally distributed for the variable “cervical range of motion” so parametric tests were used for the within (repeated measure ANOVA) and between group comparison 

Data were skewed for the variable NPRS and NDI so non parametric tests were used for the within (Friedman Test) and between (Mann Whitney U test) group comparison 

Intention to treat analysis with the technique of last observation carried forward (LOCF) was used to handle the missing data due to loss of follow-up.

Discussion:

needs to be as per well defined objectives

Describe sources of potential bias and imprecision

It has to be framed in such a way that readers are able to have good understanding of the current evidences and rationale of the paper

Discussion section has been revised.

6. PLOS authors have the option to publish the peer review history of their article (what does this mean?). If published, this will include your full peer review and any attached files.

Do you want your identity to be public for this peer review? For information about this choice, including consent withdrawal, please see our Privacy Policy.

Reviewer #1: No

Reviewer #2: No

---

## [Decision Letter · Decision Letter 1]

4 Nov 2022

PONE-D-22-23496R1Comparison of Neural Mobilization and Conservative treatment on Pain, Range of motion, and Disability in Cervical Radiculopathy: A Randomized Controlled TrialPLOS ONE

Dear Dr. Rafiq,

Thank you for submitting your manuscript to PLOS ONE. After careful consideration, we feel that it has merit but does not fully meet PLOS ONE’s publication criteria as it currently stands. Therefore, we invite you to submit a revised version of the manuscript that addresses the points raised during the review process.

We look forward to receiving your revised manuscript.

Kind regards,

Walid Kamal Abdelbasset, Ph.D.

Academic Editor

PLOS ONE

Reviewers' comments:

Reviewer's Responses to Questions

**Comments to the Author**

1. If the authors have adequately addressed your comments raised in a previous round of review and you feel that this manuscript is now acceptable for publication, you may indicate that here to bypass the “Comments to the Author” section, enter your conflict of interest statement in the “Confidential to Editor” section, and submit your "Accept" recommendation.

Reviewer #1: All comments have been addressed

Reviewer #2: All comments have been addressed

Reviewer #3: (No Response)

2. Is the manuscript technically sound, and do the data support the conclusions?

Reviewer #1: Yes

Reviewer #2: Yes

Reviewer #3: No

3. Has the statistical analysis been performed appropriately and rigorously? 

Reviewer #1: Yes

Reviewer #2: Yes

Reviewer #3: No

4. Have the authors made all data underlying the findings in their manuscript fully available?

Reviewer #1: Yes

Reviewer #2: Yes

Reviewer #3: Yes

5. Is the manuscript presented in an intelligible fashion and written in standard English?

Reviewer #1: Yes

Reviewer #2: Yes

Reviewer #3: No

6. Review Comments to the Author

Reviewer #1: Dear Author(s),

Thank you for your efforts in incorporating the improvements I proposed in the article, and best wishes.

Regards,

Reviewer #2: thanks alot for your responses

Reviewer #3: This randomized clinical trial was designed to compare the effectiveness of neural mobilization with conservative treatment on pain intensity, cervical range of motion, and disability. The conclusions are unclear.

Overall assessment: All the statistical methods used for the analysis have not been specified in the Data Analysis section. Additionally, the wording is oftentimes unclear, leading to incorrect interpretations by the reader.

Major and minor revisions:

1- Abstract: Provide a brief description of the statistical methods used to compare and express the difference within and between the groups.

2- Data Analysis section:

a. Mann-Whitney U test is misspelled.

b. This section is unclear. What comparisons were made with repeated measures ANOVA? With the Friedman tests?

3- Results section:

a. Specify the type of summary statistic that follows the +/- signs. Both variance and standard deviation are noted in the methods section.

b. Shapiro-Wilks p-values were “greater than” 0.05, would be the standard terminology.

c. Be specify, “for mostly” is too vague.

d. In the Data Analysis section, state the type of statistical method used to estimate the p-values in Tables 1 and 2. Be transparent.

e. The results show a chi-square result. List and describe the use of this method in the Data Analysis section.

f. In the Data Analysis section, list and describe all the statistical methods used.

g. Use consistent notation, for instance SD and S.D has been used. The standard is SD.

h. All acronyms and abbreviations must be spelled out at first mention.

3- Ask a native English speaker to proofread the document to improve its clarity.

4- To assist in the review process, please add line numbers to the document.

7. PLOS authors have the option to publish the peer review history of their article (what does this mean?). If published, this will include your full peer review and any attached files.

Reviewer #1: No

Reviewer #2: No

Reviewer #3: No

---

## [Author Response · Author response to Decision Letter 1]

6 Nov 2022

I am grateful to the reviewers for their insightful comments on the paper. We have been able to incorporate changes to reflect most of the reviewers’ suggestions. The text is a point-by-point response to the reviewer’s comments and concerns. 

 1-Abstract: Provide a brief description of the statistical methods used to compare and express the difference within and between the groups.

A brief description of the statistical method used has been added in the abstract section. 

2- Data Analysis section:

a. Mann-Whitney U test is misspelled.

The spelling has been corrected

b. This section is unclear. What comparisons were made with repeated measures ANOVA? With the Friedman tests?

The p-values of the Shapiro-Wilk normality test were greater than 0.05 for cervical ranges but less than 0.05 for NDI and NPRS, showing that data were normality distributed for the cervical range of motions but not normally distributed for NPRS and NDI. 

To assess the difference within the group, a parametric test i.e., repeated measure ANOVA was used for cervical ranges but its non-parametric counterpart i.e., Friedman test was used for NPRS and NDI. 

To assess the difference between the group, a parametric test i.e., independent sample t was used for cervical ranges but its non-parametric counterpart i.e., Mann-Whitney U was used for NPRS and NDI. 

3- Results section:

a. Specify the type of summary statistic that follows the +/- signs. Both variance and standard deviation are noted in the methods section.

b. Shapiro-Wilks p-values were “greater than” 0.05, would be the standard terminology.

Corrected 

c. Be specify, “for mostly” is too vague.

Corrected

d. In the Data Analysis section, state the type of statistical method used to estimate the p-values in Tables 1 and 2. Be transparent.

Table 1 and 2 shows the difference between group 1 and 2 at baseline. P values in table 1 are of the Mann Whitney U test and table 2 are of the independent sample t-test. 

e. The results show a chi-square result. List and describe the use of this method in the Data Analysis section.

The Chi-square test was not used, table 3 and 4 provide the test statistic (χ2) value ("Chi-square"), and the significance level ("Asymp. Sig."), which we need to report the result of the Friedman test.

f. In the Data Analysis section, list and describe all the statistical methods used.

All the statistical methods used are mentioned in data analysis section 

g. Use consistent notation, for instance SD and S.D has been used. The standard is SD.

Corrected 

h. All acronyms and abbreviations must be spelled out at first mention.

Corrected

---

## [Decision Letter · Decision Letter 2]

14 Nov 2022

Comparison of Neural Mobilization and Conservative treatment on Pain, Range of motion, and Disability in Cervical Radiculopathy: A Randomized Controlled Trial

PONE-D-22-23496R2

Dear Dr. Rafiq,

We’re pleased to inform you that your manuscript has been judged scientifically suitable for publication and will be formally accepted for publication once it meets all outstanding technical requirements.

Kind regards,

Walid Kamal Abdelbasset, Ph.D.

Academic Editor

PLOS ONE

Additional Editor Comments (optional):

Reviewers' comments:

Reviewer's Responses to Questions

**Comments to the Author**

1. If the authors have adequately addressed your comments raised in a previous round of review and you feel that this manuscript is now acceptable for publication, you may indicate that here to bypass the “Comments to the Author” section, enter your conflict of interest statement in the “Confidential to Editor” section, and submit your "Accept" recommendation.

Reviewer #1: All comments have been addressed

Reviewer #2: All comments have been addressed

Reviewer #3: All comments have been addressed

2. Is the manuscript technically sound, and do the data support the conclusions?

Reviewer #1: Yes

Reviewer #2: Yes

Reviewer #3: (No Response)

3. Has the statistical analysis been performed appropriately and rigorously? 

Reviewer #1: Yes

Reviewer #2: Yes

Reviewer #3: (No Response)

4. Have the authors made all data underlying the findings in their manuscript fully available?

Reviewer #1: Yes

Reviewer #2: Yes

Reviewer #3: (No Response)

5. Is the manuscript presented in an intelligible fashion and written in standard English?

Reviewer #1: Yes

Reviewer #2: Yes

Reviewer #3: (No Response)

6. Review Comments to the Author

Reviewer #1: Dear author(s),

Thank you for including the arbitrators' required amendments in the paper.

Best wishes,

Reviewer #2: thanks for your response

Reviewer #3: (No Response)

7. PLOS authors have the option to publish the peer review history of their article (what does this mean?). If published, this will include your full peer review and any attached files.

Reviewer #1: No

Reviewer #2: No

Reviewer #3: No

---

## [Editor Report · Acceptance letter]

23 Nov 2022

PONE-D-22-23496R2 

Comparison of Neural Mobilization and Conservative treatment on Pain, Range of motion, and Disability in Cervical Radiculopathy: A Randomized Controlled Trial 

Dear Dr. Rafiq:

I'm pleased to inform you that your manuscript has been deemed suitable for publication in PLOS ONE. Congratulations! Your manuscript is now with our production department. 

Kind regards, 

on behalf of

Dr. Walid Kamal Abdelbasset 

Academic Editor

PLOS ONE